# Operando Li metal plating diagnostics via MHz band electromagnetics

**Masanori Ishigaki** [1] ✉, **Keisuke Ishikawa** [1] ✉, **Tsukasa Usuki**[1], **Hiroki Kondo** [1], **Shogo Komagata**[1] & **Tsuyoshi Sasaki**[1]

A nondestructive detection method for internal Li-metal plating in lithium-ion batteries is essential to improve their lifetime. Here, we demonstrate a direct Li-metal detection technology that focuses on electromagnetic behaviour. Through an interdisciplinary approach combining the ionic behaviour of electrochemical reactions at the negative electrode and the electromagnetic behaviour of electrons based on Maxwell's equations, we find that internal Li-metal plating can be detected by the decrease in real part of the impedance at high-frequency. This finding enables simpler diagnostics when compared to data-driven analysis because we can correlate a direct response from the electronic behaviour to the metallic material property rather changes in the ionic behaviour. We test this response using commercial Li-ion batteries subject to extremely fast charging conditions to induce Li-metal plating. From this, we develop a battery sensor that detects and monitors the cycle-by-cycle growth of Li-metal plating. This work not only contributes to advancing future Li-ion battery development but may also serve as a tool for Li-metal plating monitoring in real-field applications to increase the useable lifetime of Li-ion batteries and to prevent detrimental Li-metal plating.

The use of lithium-ion batteries (LiBs) is projected to increase by a factor of 10–20 in the next decade[1]. For this growth to be sustained, the improvement of the battery performance and the resource availability of Co, Ni, and Cu[2–4] should be considered along with the life cycle assessment (LCA) for reducing the $CO_2$ emissions from production to disposal[5,6]. Therefore, technologies to improve the service life of LiBs are considered to be environmentally beneficial[7].

The challenges of long-use in LiBs have been highlighted in recent reuse concepts, such as electric vehicle batteries in renewable energy storage stations in the electric power grid[8,9]. In one example, diagnostic technologies are required for detecting its continuous value. The concept of electrochemical impedance spectroscopy (EIS), which nondestructively estimates internal conditions from the inherent response of materials, is widely used in various scientific fields as a diagnostic technology[10]. Batteries are one such example, where EIS is a well-known tool to diagnose the state of health (SOH) that indicates the remaining power and capacity performance via analysis of the mHz-to-kHz frequency response[11–15]. To evaluate the residual value of a LiB, the safety level should be first guaranteed in accordance with regulatory standards by battery manufacturers, as long as LiBs are categorised as inherently flammable devices[16–20].

Monitoring the internal growth of Li-metal deposition is critical for evaluating state of safety (SOS) of the LiBs, because such growth has been identified as a factor of reducing thermal runaway temperature of the LiBs[21–27]. Therefore, each new battery system must include its respective charge/discharge control scheme to prevent Li-metal deposition. The battery safety is a concern not only for battery application users but also for ancillary industries such as transportation and storage for battery reusing and recycling[28,29], because such industries cannot evaluate the internal Li-metal deposition by destructive processes[8,9]. Hence, a reliable, convenient and accurate technology for detecting internal Li-metal deposition would facilitate the social circulation of batteries[30].

[1]Secondary Batteries Research-Domain, Toyota Central R&D Labs., INC, Nagakute, Japan. ✉e-mail: ishigaki@mosk.tytlabs.co.jp; k-ishikawa@mosk.tytlabs.co.jp

In this study, we explore an innovative safety diagnostic technology that does not rely on physical examination or usage history. We have developed a direct metal detection mechanism by utilising a high-frequency electromagnetic response in LiB, demonstrating an approach that has received little attention in electrochemical devices. In this mechanism, Li-metal plating can be distinguished by a measurable change in the real part of the impedance in the specified high-frequency (MHz) band, separate from other degradation factors such as the growth of the solid−electrolyte interphase (SEI) or cathode collapse[31]. This discovery is easily applied as a nondestructive, quick measurement that cannot otherwise be achieved by conventional nondestructive, direct diagnostic tools such as neutron and muon quantum beams that require specialised or large-scale approaches[32,33].

## Results

### High-frequency electromagnetic behaviour in Li-metal deposited LiB

Our experiments clearly detect the existence of Li-metal plating based on the response of the high-frequency electronic current, which directly reacts to changes in the conductivity of the material in accordance with Maxwell's equations. This mechanism is fundamentally different from those usually attained by electrochemical methods, which diagnose LiB safety based on a degradation in ionic reaction. This mechanism is also different from conventional magnetohydrodynamic effects, which have been studied as the reaction of

charged ions in LiB by DC and low-frequency magnetic fields[34–36]. Here, the detection mechanism is attributed to the extended equivalent circuit model (ECM) shown in Figs. 1, 2, with the corresponding electromagnetic simulation result for the high-frequency EIS shown in Fig. 3. Figures 1, 2 compares the high-frequency electromagnetic response to those of low-frequency ECM. In electrochemistry, Li-metal plating degrades its output performance by interfering with lithium intercalation as well as safety degradation (Fig. 1a)[37–39]. The ECM shown in Fig. 1b, c simulates the battery response at low-frequency bands (DC to kHz). The reaction of the Li-ion battery is simply described by a multiple $RC$ circuit 1D model shown in Fig. 1b and refs. 40,41. $RC$-Model A represents the double-layer reaction, and $RC$-Model B represents the SEI response on the negative electrode surface. Generally, $RC$-Model A transitions from a resistive response described as a diffusion resistance and a charge transfer resistance $Z_w + R_{ct}$ to a capacitive response described as a double layer capacitance $C_{dl}$ at frequencies above 10 Hz. As an extension of the 1D model, a distributed behaviour model shown in Fig. 1c can be used to describe the diffusion of the lithium-ion along the stack direction[42,43]. In this model, the resistance of the Li-metal plating $R_{Li-M}$ is represented as a slight increase in the series-connected resistance to the boundary of the graphite $R_{gr}$, the SEI resistance $R_{SEI}$, and electrolyte impedance $Z_{elec}$ (Supplementary Fig. 1a–c). Thus, Li-metal plating is detected along with other degradation factors as impedance changes in conventional low-frequency EIS (Supplementary Fig. 1d, e). Although some prior studies have explored the

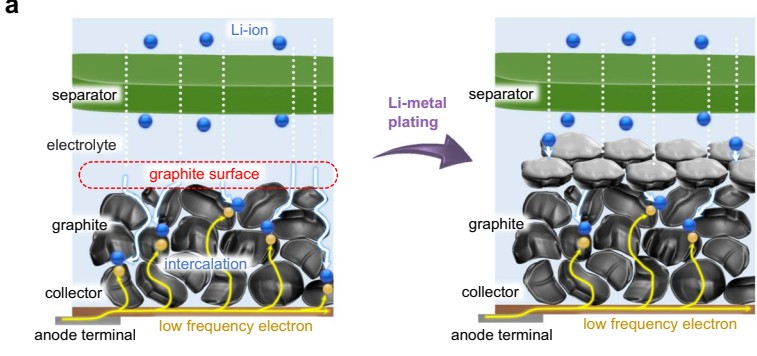

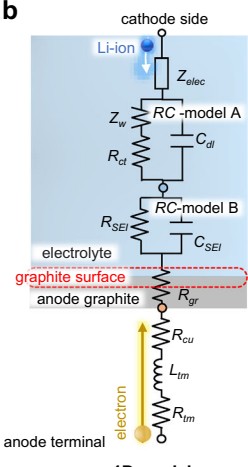

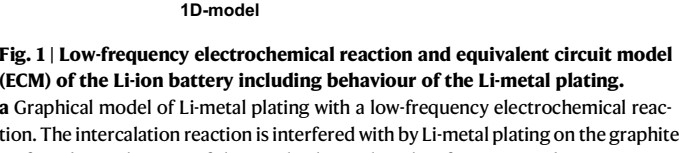

**1D-model**

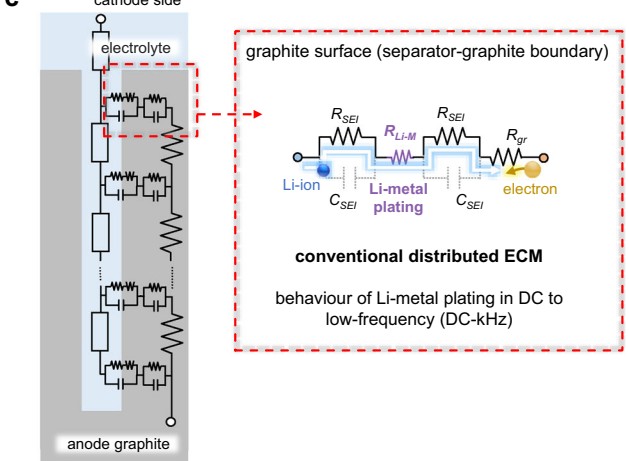

**Fig. 1 | Low-frequency electrochemical reaction and equivalent circuit model (ECM) of the Li-ion battery including behaviour of the Li-metal plating.** **a** Graphical model of Li-metal plating with a low-frequency electrochemical reaction. The intercalation reaction is interfered with by Li-metal plating on the graphite surface. **b**, **c** Behaviour of the anode electrode at low frequencies. **b** One-dimensional (1D) ECM of the anode electrode with frequency-dependent behaviours. $R_{tm}$ and $L_{tm}$ represent terminal impedance, and $R_{cu}$ and $R_{gr}$ represent

resistance of the anode collector and the active material. $RC$-Model A consisting of $C_{dl}$, $R_{ct}$ and $Z_w$ describes the charge transfer impedance, and $RC$-Model B consisting of $C_{SEI}$ and $R_{SEI}$ describes the impedance of the SEI layer. **c** Conventional distributed ECM that can model kHz band frequency dependence in the stacking direction. The resistance of Li metal $R_{Li-M}$ appears as a series resistance to $RC$-Model B which is located on the surface side of the $RC$ ladder network.

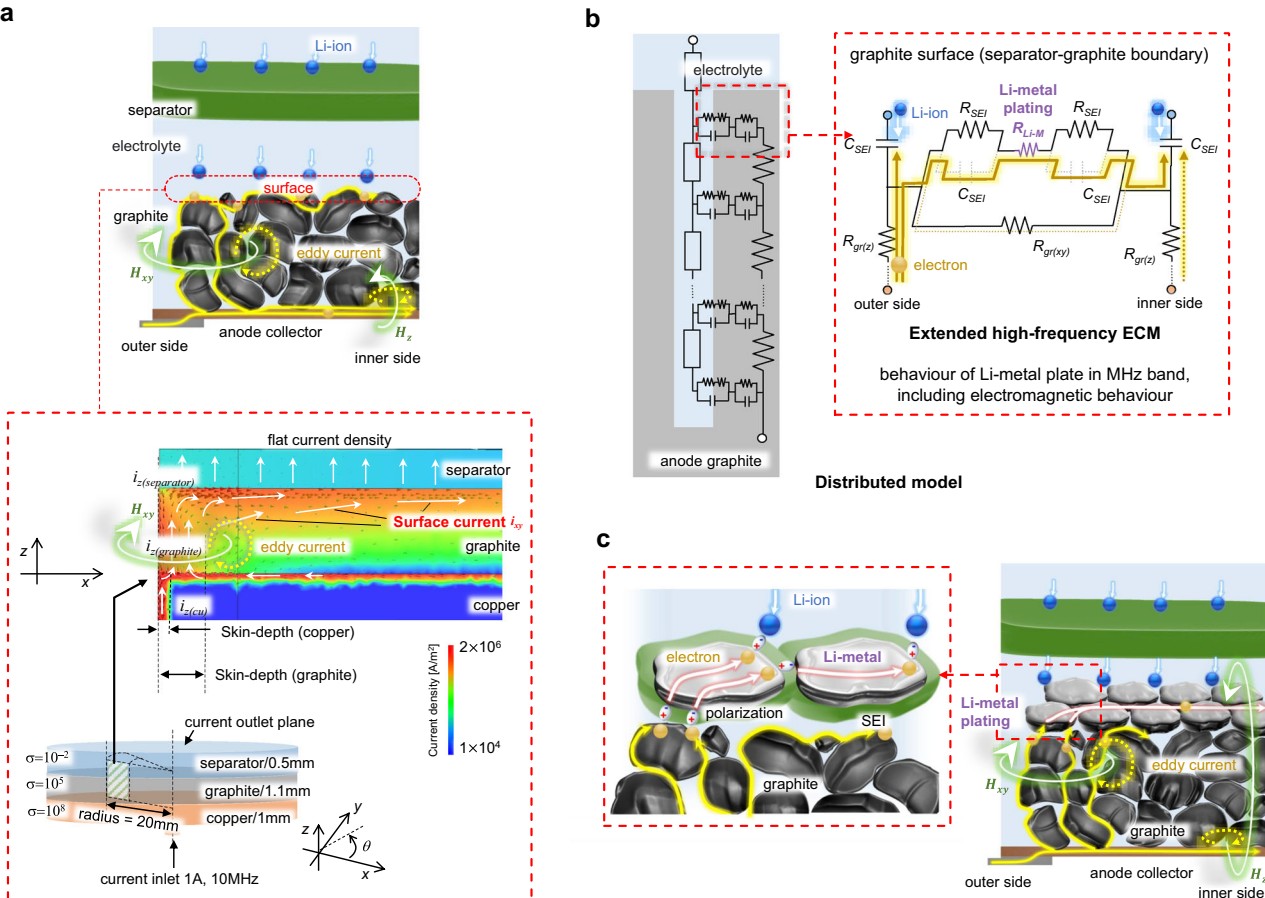

**Fig. 2 | High-frequency response of the Li-ion battery including electro-magnetic response. a** Graphical model and simplified electromagnetic FEM analysis of MHz band behaviour. Both model and analysis display current concentration by the skin effect in multilayered, different conductive materials. The counter of the current density in the $xy$ surface of the edge of the cylinder model visualises the current distribution in the $z$ and $xy$ directions by electromagnetic FEM analysis. **b** Extended high-frequency ECM of the anode electrode with electromagnetic behaviours. In MHz, the $RC$-Model B including $R_{Li-M}$ is located parallel to $R_{gr(xy)}$, which describes graphite resistance in the $xy$ direction. Therefore, the $R_{Li-M}$ would appear as a low-resistance channel against the surface current in the $xy$ direction. **c** Graphical model of Li-metal plating with a MHz electromagnetic reaction. The electron current distributes to the $xy$ direction on the graphite surface where the Li-metal plate grows. This surface current can pass through Li-metal via a polarisation layer ($C_{SEI}$) as a displacement current.

application of machine learning to achieve factor separation, a significant amount of empirical data is required to distinguish Li-metal plating from the other major degradation factors since the ionic response in the low-frequency impedance is dependent on the material properties[44–47]. In contrast, at frequencies above 100 kHz, the electromagnetic effects based on Maxwell's equations become significant and drive the current concentration in the conductive layers. Figure 2 describes the high-frequency electromagnetic behaviour of the LiB with extended high-frequency ECM. The feature of the current concentration can be explained using a simplified FEM (finite element method) analysis, as shown in Fig. 2a. When applying a z-direction current with frequency $f_r$ to a conductive material, the magnetic field $H_{xy}$ induces a current concentration as $i_z$ near the thickness $\delta_{(\sigma)} = (\pi f_r \mu \sigma)^{-0.5}$ from the edge of the conductive material, which has conductivity $\sigma$ and permeability $\mu$. This effect is called as skin effect, which varies the $\delta_{(\sigma)}$ arising in each material with different conductivities. Therefore, a surface current $i_{xy}$ emerges on the material boundary to mitigate the difference in current density $i_z$ (see Supplementary Discussion 1)[48]. To describe this current distribution on the graphite surface, the high-frequency behaviour model shown in Fig. 2b decomposes the graphite resistance into two components: $R_{gr(z)}$ in the stacked direction and $R_{gr(xy)}$ in the surface direction. Hence, the Li-metal plate is deposited on the graphite surface, and $R_{Li-M}$ with $RC$-Model B is connected in parallel to $R_{gr(xy)}$. Figure 2c graphically

summarises the features of the high-frequency current distribution. The electron current passes through the edge of graphite at high-frequency, including the boundary between the graphite and the separator as the $i_{xy}$. The Li-metal plating was deposited on the way of $i_{xy}$. The $i_{xy}$ can trough the Li-metal plating even though it forms the SEI, since the SEI model shown as $RC$-Model B could transform its reaction from resistive $R_{SEI}$ to capacitive $C_{SEI}$ in the MHz frequency band. As a result, the $i_{xy}$ flow is concentrated on the thin metal surface by the growth of Li-metal plating, which has a larger conductivity than graphite.

The magnetic field $H_z$ caused by the $i_{xy}$ represents a source of Li-metal detection. Figure 3 demonstrates the response of the real part of the impedance against Li-metal plating based on electromagnetic FEM analysis. The high-frequency magnetic field $H_z$ induces another current concentration against the $i_{xy}$ that diffuses on the graphite surface. This effect is called the proximity effect, which attracts the loop current of $H_z$ into the inside path by the Lorentz force. As shown in Fig. 3a, by the frequency increase, the $i_{xy}$ that diffuses within the metal plate at 1 kHz is concentrated inside the current loop by the proximity effect at 10 MHz. The mechanism of the high-frequency EIS is explained by using a 3D stacked layer model where the material conductivity of the graphite surface layers changes its properties to those of Li metal (Fig. 3b and Supplementary Fig. 2a–c). Figure 3c–e and Supplementary Fig. 2d–h show the FEM analysis results at 10 MHz. Figure 3c illustrates

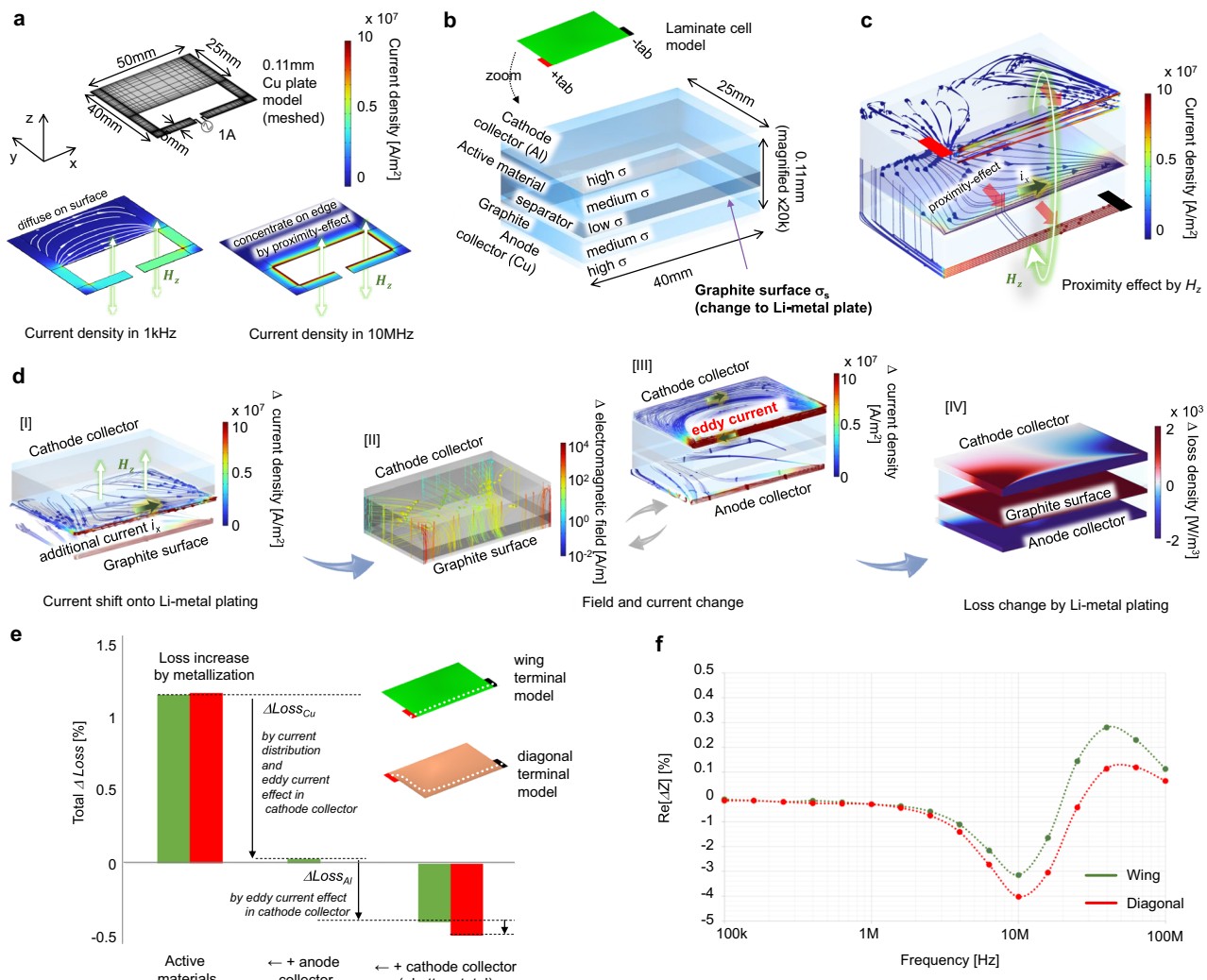

**Fig. 3 | Detection mechanism of Li-metal plating through high-frequency electromagnetics. a** Current concentration by the proximity effect. The copper model imitates the laminate cell shown in (**b**) and the Supplementary Figure 3. The vector line shows the current distribution in the *xy* plane. **b**–**e** Results of electromagnetic FEM analysis at 10 MHz. A current of 1 A is input from the anode and cathode tab. **b** Electromagnetic model of the laminate cell for visualising the high-frequency behaviour of the electron. The stacked direction (z) is magnified 20k times to visualise the high-frequency current distribution in the thin layers. **c** Overall current distribution at 10 MHz. The coloured vector line shows the intensity and 3D behaviour of the high-frequency current highlighting the

anode–separator boundary, cathode collector, and anode collector. **d** Electromagnetic response against Li-metal plating. The coloured vectors show the variation in the current density in [I] and [III], and the electromagnetic field in [II]. In this process, [II] and [III] are convergent behaviours. The Δ current density in [III] drives the loss change as described by the red and blue counter in [IV]. **e** Elements of the loss change categorised as active materials, anode collector, and cathode collector. The tab design is arranged in two models to clarify the effect of $H_z$. A negative change in the total loss can be measured as a decrease in resistance Re[Z]. **f** Result of the electromagnetic FEM analysis over a wide frequency range. Re[ΔZ] will appear as a measurable change in the terminal impedance.

the 3D current distribution with the skin effect and the proximity effect. As shown therein, when a high-frequency current is applied to a battery model from the tabs at both ends in the *x* direction, the proximity effect concentrates the $i_{xy}$ on the graphite edge as $i_x$ in the longitudinal direction of the battery. As a result, a penetrating magnetic field $H_z$ is generated inside the laminated battery from the anode to the cathode. In addition, $H_z$ induces loss changes in the metal objects in the penetrating path, similar to an induction metal detector[49]. Figure 3d illustrates the mechanism of loss changes in the battery model by $H_z$. The Li-metal plating enhances the increases of the edge current $i_x$ by higher material conductivity on the anode surface [I]. Consequently, Li-metal plating increases the intensity of the $H_z$ that penetrates the anode to cathode layers [II]. Simultaneously, magnetic flux linkage by $H_z$ generates eddy current in the cathode collector [III]. As a result, Li-metal plating induces a loss change on the

cathode side [IV]. The numerical results of the FEM analysis are summarised in Fig. 3e, f. The change in measurable real part of the impedance Re[Z] appears to be negatively correlated with the amount of Li-metal plating. Figure 3e summarises the factors of loss variation in the FEM analysis. The metallisation of the graphite surface causes an additional surface diffusion current in the anode active material, which migrates from the copper collector. Therefore, Li-metal plating increases the loss in the active material, while the loss in the copper collector decreases. The balance between those increases and decreases is cancelled in the anode and therefore measured as approximately zero. The total loss change becomes negative by the cathode collector because the eddy current in the cathode collector mitigates the high-frequency current concentration, as shown in Fig. 3d [IV]. This loss change was measured as the change in real part of the impedance Re[ΔZ]. Figure 3f summarises the frequency response

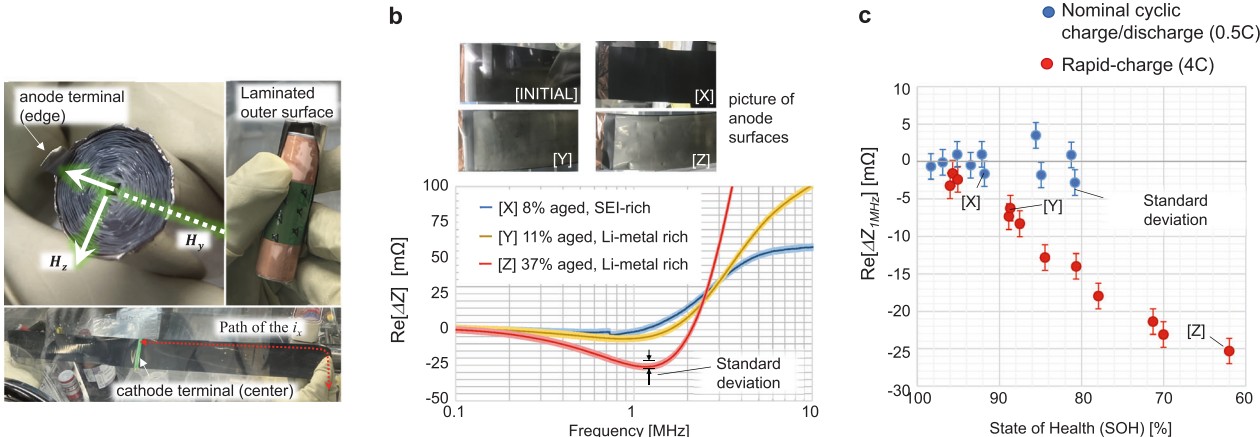

**Fig. 4 | Measured high-frequency impedance in a commercial 18,650-type battery (1500 mAh, LFP). a** Internal structure of the 18,650-type battery used in this verification. This rolled electrode structure generates an additional high-frequency magnetic field $H_y$ that penetrates in the ±terminal direction of the battery. However, this additional magnetic field has less interference against the Li-metal detection mechanism since $H_y$ is perpendicularly across against the $H_z$. **b, c** Measured battery impedance change Re[$\Delta Z$] at 20 °C and SOC = 60%. The standard deviation at 1 MHz is ± $(1.61^2 + 1.612)^{0.5}$ = ± 2.27 mΩ, based on the Supplementary Fig. 6. The blue line and dots are degraded by cyclic 0.5 C charge and discharge at 60 °C, and the red lines and dots are degraded by 4 C rapid charge at 20 °C. **b** Frequency response of the measured Re[$\Delta Z$] in degraded batteries, with anode surface verification by destructive observation. The SEI-rich battery [X] degraded its capacity by 8% from the initial value. The black anode surface in [X] is the colour of the graphite, which is the same colour as the INITIAL. [Y] and [Z] degraded 11% and 37% of the SOH by 4 C rapid charging. The silver colour shown in the surface observation of [Y] and [Z] is the Li-metal plating. **c** SOH versus Re[$\Delta Z$] measured at 1 MHz. Twenty-one individual batteries are measured by controlling different SOH degradations by arranging the number of charging cycles. The result of the single-frequency measurement of Re[$\Delta Z_{1MHz}$] successfully categorises two different conditions of the degraded batteries.

of Re[$\Delta Z$] in the laminate models. The negative correlation reaches a maximum at 10 MHz. The intensity of the negative correlation increases when the terminals are located on the far side since it drives $H_z$ stronger by a longer edge for the $i_x$, such as the difference between wing terminal mode to diagonal terminal model in Fig. 3e.

In this electromagnetic FEM analysis, all the ionic behaviour is zeroed out by approximating impedance of the electrolyte $Z_{elec}$ as a linear dielectric solid layer with leak resistance similar to a dielectric layer of a capacitor. In actual batteries, the decrease in ionic resistance is dependent on the material properties of the dielectric layers, which can be described as high-frequency characteristics of the RC-model B. In addition, in such high-frequency, the increase in electronic resistance of the graphite layer depends on the electrode structure as shown in Figs. 2, 3. Therefore, transition frequency from the conventional ECM to the extended high-frequency ECM can vary in each individual battery design. In this study, within the frequency range where the real part of the impedance is measured as clear increase against frequency, we consider the behavior of the battery can be enough described as the extended high-frequency ECM for Li-metal detecting. Moreover, the RC parallel circuit behaves more capacitive by increasing the surface area, then the electromagnetic behaves gets majority in the real part of the impedance in high-frequency. Therefore, this Li-metal detection method works better with large capacity batteries. In addition, this finding has less dependence on the variety of cathode materials, since the electronic behaves is the measuring object. However, its results may be affected by surrounding metal changes, e.g., deformation of metal enclosures. Extensive quantitative analysis using laminate-type pouch Li-metal deposited cells (Supplementary Fig. 3) and destructive analysis for Li-metal quantification experimentally verified the negative correlation between Li-metal plating and high-frequency impedance. Due to the small capacitance in $Z_{elec}$ of the pouch cell, the ionic behaviour is not zeroed out for directly sensing the high-frequency response from Li-metal plating (Supplementary Fig. 1b, c). Therefore, conventional EIS modelling is used to derive the negative correlation between Li-metal plating and high-frequency impedance from the experimental result (Supplementary Fig. 3c–f). These results also verify that the impedance change

responds to the average conductivity of the anode surface, whereas the Li-metal plating is speckled.

## Nondestructive detection of Li-metal plating in commercially available batteries

In addition to revealing a unique response of high-frequency electromagnetics, we propose that our discovery represents an important engineering innovation because it pertains to both battery development and practical battery usage scenarios.

Figures 4 and 5 present the results of the safety diagnosis using the proposed Li-metal plating detection. By observing Li-metal deposition in 18,650 commercially available batteries (Supplementary Figs. 4–6), we demonstrate the development of a nondestructive, operando battery safety diagnostic system. This result is significant for three reasons: it can be used for practical battery sizes, it can identify Li-metal plated aging within the same range capacity as a degraded battery, and it can be integrated into a simple monitoring sensor for operando diagnostics. Additional results of our experiments, such as material compatibility or structural dependency, are presented in Supplementary Fig. 5. In order to ensure the accuracy of Re[$\Delta Z$], the variability of contact resistance and the stray loss occurred in surrounding metal need to be carefully minimized. Supplementary Fig. 6a summaries practical accuracy of the measurement method as standard deviation.

Figure 4 summarises the measurement results of the high-frequency impedance in an 18,650-type battery (LFP, 1500 mAh) using a vector network analyser (VNA). In real battery structure, an additional high-frequency magnetic field $H_y$ is induced in most practical batteries by its rolled structure (Fig. 4a), and $H_y$ exhibits less interference against the Li-metal detection mechanism since $H_y$ crosses perpendicularly against $H_z$. Moreover, $H_y$ may enhance the detection sensitivity against the growth of Li-metal plating by concentrating the high-frequency current on the surface of the cylindrically oriented battery structure, which typically starts Li-metal plating in a cold environment due to poor self-heating.

The overall frequency response of Re[$\Delta Z$] is shown in Fig. 4b. In this result, a significant decrease in Re[$Z$] is confirmed in the range of

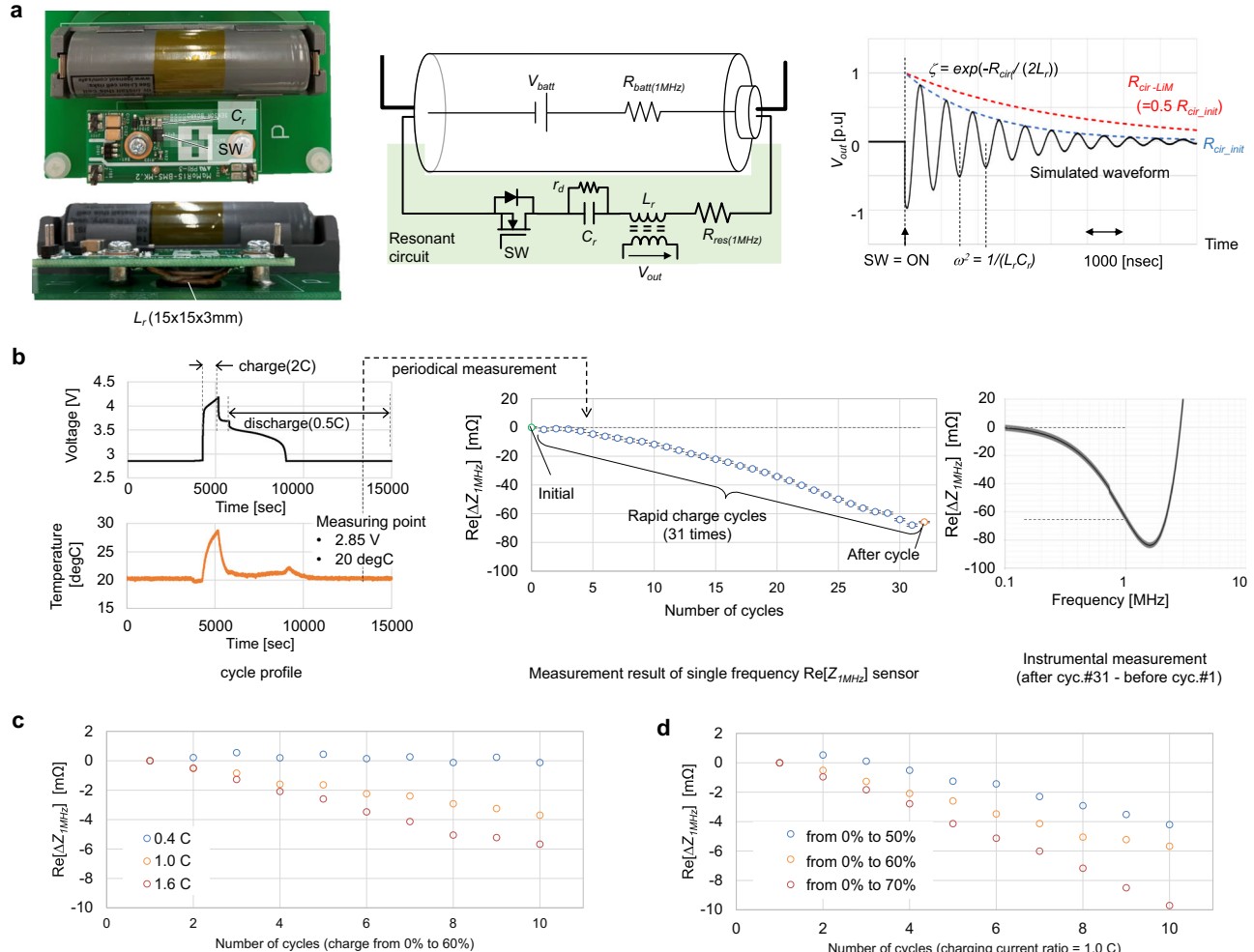

**Fig. 5 | Demonstration of safety diagnosis in a commercial 18,650 battery (2600 mAh, NCM). a** Picture, circuit diagram and basic operation waveform of prototype sensor for operando safety monitoring. The sensor consists of a series LC resonant circuit and a switch (SW), and directly connected to the battery via a printed circuit board. The resonant frequency of the LC resonance is set as approximately 1 MHz by $L_r$ = 1000 nH and $C_r$ = 27 nF (nominal values). $R_{batt\,(1\,MHz)}$ represents the battery resistance at 1 MHz. $R_{res\,(1\,MHz)}$ represents the total resistance of the circuit, including the circuit pattern, the LC components, the SW, and contact resistance of the battery holder. The inductor $L_r$ has a secondary winding that can pick up the resonant waveform as $V_{out}$ with isolation. After turning on the SW, the circuit starts dumping oscillation, as shown in the simulated waveform. The dumping factor ζ is a function of $R_{cir}$ = $R_{res(1\,MHz)}$ + $R_{batt(1MHz)}$, which represents the overall series resistance of the resonant loop. Re[$\Delta Z_{1MHz}$] is the same as the change in $R_{batt(1MHz)}$, where $R_{res(1MHz)}$ is constant. Re[$\Delta Z_{1MHz}$] can be calculated by the change in ζ. Parallel connected resistance $r_d$ discharges $C_r$ after resonance. **b** Tracking result of Re[$\Delta Z_{1MHz}$] during the 2 C rapid charged ageing test. The result shows the change in Re[$\Delta Z_{1MHz}$] cycle by cycle by measuring the sensor output in the steady state, 2.85 V during a 0.5 C discharge, at 20 °C. The measurement error shown in Supplementary Fig. 6d is described as an error bar. The final result of Re[$\Delta Z_{1MHz}$] is measured by the VNA to verify the sensor result. **c, d** Monitoring result of accumulation of the Li-metal plating in multiple charging conditions. The basic degradation pattern and measuring point are the same as in (**b**). **c** Variation in Re[$\Delta Z_{1MHz}$] vs. number of cycles, with the C-ratio as a variable. The ΔSOC is set as 60%, and the start SOC is set as 0%. **d** Variation in Re[$\Delta Z_{1MHz}$] vs. number of cycles, with ΔSOC as a variable. The C-ratio is set as 1 C.

500 kHz to 2 MHz against Li-metal plating. After specifying the sensitive frequency $f_m$, the presence of Li-metal plating is classified by tracking a change in the specified frequency impedance Re[$\Delta Z_{fm}$], as shown in Fig. 4c. In the experimental battery, Re[$\Delta Z_{1MHz}$] clearly shows two types of aged batteries: red-coloured batteries that are intentionally subjected to excessive rapid charging to promote Li-metal plating and blue-coloured batteries that age normally due to use within rated charging and discharging specifications. The response of Re[$\Delta Z_{1MHz}$] to the amount of Li-metal plating is similar to the negative correlation results in Supplementary Fig. 3g. The sensitive frequency range is dependent on the structural feature of each battery, such as the effect of $Z_{elec}$ shown in Supplementary Fig. 1.

Figure 5 and the Supplementary Fig. 6b–d demonstrate the operando monitoring results of the growth of Li-metal plating by applying a simple prototyped LC resonant circuit that can track the change in Re[$\Delta Z_{fm}$]. After specifying the frequency $f_m$ and variation range of Re[$\Delta Z_{fm}$] for each battery, battery safety monitoring is achieved throughout the operational lifetime by tracking the change in Re[$\Delta Z_{fm}$]2. The circuit shown in Fig. 5a is designed to track the variation in the resonant loop resistance $R_{cir}$, which includes Re[$\Delta Z_{1MHz}$], from the change in the dumping envelope of the resonance. The series resonant topology enables the sensor circuit to be connected directly to the battery without interfering with the main charge/discharge current. The tracking result of Re[$\Delta Z_{1MHz}$] during 2 C rapid charging is summarised in Fig. 5b. The sensor output of each cycle can detect the gradual decrease in Re[$\Delta Z_{1MHz}$]. The final result of Re[$\Delta Z_{1MHz}$] confirms accurate tracking by referring to the instrumental measurement method shown in Fig. 4. This result confirms our finding that continuous monitoring of the accumulation of Li-metal plating is enabled. Therefore, we significantly contribute to long-life battery utilisation by

monitoring its safety level degradation. Our operando monitoring is also expected to contribute to accelerating battery system development by enabling immediate observation of battery degradation under different conditions. As shown in Fig. 5c, d, the sensor result can nondestructively detect the accumulation of Li-metal plating under different charging conditions within a few accelerating cycles. These results are in line with the deposition principle in LiB, where a higher charge current and wider charge depth result in greater deposition of Li-metal plating[23,26,37,50,51].

## Discussion

This study bridges electrochemical and electromagnetic approaches to testing based on a simple mechanism of monitoring the high-frequency electron behaviour in the battery. Our results indicate that additional research in this field has the potential to solve or reveal other unexplored battery characteristics. For example, this operando technique be a strong tool for accelerating the battery development of lithium metal batteries or all solid batteries by monitoring Li-metal deposition without a destructive approach[52,53]. Moreover, throughout our experiments, $\text{Re}[\Delta Z]$ data in tens of MHz show a strong correlation with capacity degradation (Supplementary Fig. 4), revealing the potential of SOH estimation via high-frequency impedance measurement. Although the high-frequency characteristics verification for the SEI[54,55] present another challenge, this approach will demonstrate high-frequency measurements to be an even more powerful tool for battery development and diagnostics in this important research field.

Finally, by combining the above development method and diagnostic potential, we propose that this technology can contribute greatly to the long-life use of batteries and help advance their carbon neutrality.

## Methods

### High-frequency (100 kHz–10 MHz) impedance measurement

The high-frequency impedance is measured using a two-port vector network analyser (Keysight VNA E5061B, ±7 V input, 100 kHz to 1.5 GHz) and the shunt-through measurement technique that is most accurate method for low resistance measurement (averaged eight times). Two calibrated SMA coaxial cables (less than 2.0 m, calibrated by Keysight N7555A) are connected from the instrument to the SMA connector on the battery side, which is placed in the constant-temperature chamber. The battery is inserted in a plate-spring holder connected to SMA connectors on an FR4 printed circuit board (PCB). The expression for the two-port S-parameter with a shunt-through connection is as follows:

$$\begin{bmatrix} S_{11} & S_{12} \\ S_{21} & S_{22} \end{bmatrix} = \frac{1}{\frac{Z_0}{2} + Z_L} \begin{bmatrix} -\frac{Z_0}{2} & Z_{bat} \\ Z_{bat} & -\frac{Z_0}{2} \end{bmatrix},$$

where $Z_0$ is the VNA's reference port impedance and $Z_{bat}$ is the battery impedance. The measured S-parameter values are converted to the impedance values by the following equation:

$$Z_{bat} = \frac{\frac{Z_0}{2} S_{21}}{1 - S_{21}}$$

Based on the quarter wavelength of the coaxial cables (37 MHz in a 2 m cable), the precise frequency range is 20 MHz. There is stepwise fluctuation at 733 kHz due to instrument features. In this setup, with a frequency range of 100 kHz–20 MHz, the standard deviation of the real part of the 18,650 battery impedance was less than 1.61 mΩ and the coefficient of variation was less than 0.37 % (1 MHz–20 MHz).

### Electromagnetic simulation

The electromagnetic simulation was analysed using COMSOL 6.0 with varying conductivities in multiple layers, corresponding to the

laminate model shown in the Supplementary Fig. 1. Considering the high-frequency characteristics presented in Fig. 2, the ionic response is set to approximately 0 by describing the electrolyte–separator model component as having an extremely low fixed conductivity. To produce appropriate high-frequency characteristics, the separator layer is constructed to have a dielectric permittivity depending on the material property for emulating the dielectric polarisation. In this analysis, the COMSOL model utilized RF module which can calculate impedance of the model. In this model, lumped ports were installed in predetermined locations, and loss and the real part of impedance were determined by inducing a 1 A constant current. Considering the computational cost, the mesh is made fine enough to represent the skin effect at the edges, the other battery-simulated parts are unified with a rectangular mesh, and the remaining sections, such as terminals, are unified with a triangular mesh. To display the results, the software magnifies the stacking direction to make the representation of the magnetic field easier to comprehend.

### Battery selection and degradation conditioning of the 18,650 cells

The performance of commercially available LiBs was examined. There are variants such as the Panasonic NCR18650b (3350 mAh, 3.6 V), LG INR18650 M26 (2600 mAh, 3.65 V) and O'CELL IFR18650EC-1.5 Ah (1500 mAh, 3.2 V). The positive electrode compositions of NCR18650b (NCA), INR18650 M26 (NCM) and IFR18650EC-1.5 Ah (LFP) were $LiNi_{0.8}Co_{0.15}Al_{0.05}O_2$, $LiNi_{0.49}Co_{0.2}Mn_{0.31}O_2$ and $LiFePO_4$, respectively. Supplementary Table 1 provides additional information on the batteries. The Li-metal deposited degradation of the NCA, NCM and LFP batteries was evaluated using excessive rapid charge cycles. The results were examined using a VNA. The excessive rapid charge cycle test consisted of a constant current (CC) charge (at 2 C, 2 C, and 4 C; cut-off voltages of 4.2 V and 3.9 V, respectively); 10 min rest at 20 °C; slow constant current constant voltage (CCCV) discharge (at 0.1 C and 3 h; cut-off voltages of 2.5 V and 2.0 V, respectively); and 10 min rest at 20 °C. A high-temperature charge cycle test for the non-Li deposition deterioration was performed on both the NCA and LFP batteries. The results were examined using a VNA. The non-Li deposition degradation test was cycled between a slow CC charge (at 0.5 C, cut-offs of 4.2 V and 3.65 V), a rest time of 10 min at 60 °C, and a slow CC discharge (at 0.5 C, cut-offs of 2.5 V and 2.0 V). Using the sensor, an NCM battery was subjected to an excessive rapid charge cycle test for Li-metal deposition, and the data obtained were analysed. The Li deposition degradation tests consisted of an excessive rapid CC charge (at 2 C, 4.3 V cut-off), 10 min of rest, slow CCCV discharge (at 0.5 C, 3 h, 2.85 V cut-off) and 10 min of rest at 20 °C. Supplementary Table 2 provides the additional experimental parameters.

### Materials of the laminate-type pouch cell

Hosen Corporation supplied the positive electrode consisting of $LiNi_{1/3}Mn_{1/3}Co_{1/3}O_2$ (NMC111) and a negative electrode consisting of spherulite graphite (TSG-A1). Before the fabrication of the laminate-type pouch cell with a 14 mAh capacity, both electrodes were cut to a size of 25 × 38 mm (NMC111) or 25 × 40 mm (TSG-A1) and dried at 80 °C under vacuum for 10 h. The electrolyte (1.1 M $LiPF_6$ dissolved in a 30:40:30 volume ratio of ethylene carbonate, dimethyl carbonate, and ethyl methyl carbonate) was purchased from KISHIDA CHEMICAL Co., Ltd. Toray Industries, Inc., which also supplied the microporous polyethylene film (29 × 65 × 0.02 mm) that was used as a separator.

### Fabrication, degradation, and characterisation of the laminate-type pouch cell

Laminate-type pouch cells were fabricated by combining the materials described above. The positive and negative electrodes, separator, and electrolyte were assembled in an argon-filled glovebox. The amount of electrolyte contained in the battery was adjusted through vacuum

impregnation, rather than based on predefined liquid volume specifications. After removing the batteries from the glovebox during the electrolyte injection, the impedance measurements were made by connecting a custom PCB in series (Supplementary Fig. 3a). The cells were charged and discharged twice with a CC and a constant voltage (3.0–4.1 V) at 20 °C and 0.2 C and voltage holding for 2 h, where 1 C is set as 14 mA. The cells were stored at 60 °C for 12 h. Li metal was deposited by cycling the batteries for 10 and 20 cycles at 20 °C with charging and discharging currents at 4 C. After the cycling test, one of the cells after 10 cycled was stored at 60 °C for 7 days. The initial and degraded cells were adjusted to 3.65 V, with 40% state of charge (SOC).

The high-frequency impedance and low-frequency impedance of the fabricated cells were measured. A high-frequency impedance evaluation was conducted as mentioned above. At 20 °C, the impedance was measured from 10 mHz to 10 kHz with a voltage amplitude of 10 mV. The anode sheets were removed from the cells, washed with solvent, and dried in an argon-filled glovebox (Supplementary Fig. 3b) after the impedance measurements. These anode sheets were divided into two parts for inductively coupled plasma–optical emission spectrometry (ICP–OES) and solid [7]Li nuclear magnetic resonance (NMR) measurements conducted at the Toray Research Centre. Using ICP–OES, the total amount of all the Li chemical species in the anode was detected. Solid-state [7]Li NMR was used to determine the composition ratio of the Li chemical species. The amount of Li metal was estimated using the results from the ICP–OES and solid [7]Li NMR measurements.

## Extraction of the electronic impedance of the laminate-type pouch cell in a high-frequency with the conventional ECM

The conventional ECM (Supplementary Fig. 3c) was used to fit the Cole-Cole plot of the measured impedance in Supplementary Fig. 3f by using Impedance.py (1.4.0). The starting frequency (100 mHz) is set by the measured Cole-Cole plot, where the diffusion impedance could be regarded as a negligible value. A value of 750 kHz (-Im[$Z$] > 0) was set as the end point of the fitting frequency, where the value of the Re[$Z$] was conventionally treated as an ohmic resistance in the Cole-Cole plot. When the frequency range exceeded 1 MHz, the measured curve did not correspond to the computed curve. This variation accurately represents the impedance at high frequencies that cannot be described using the usual ECM formula. By subtracting the extended fitting curve from the measured curve, the impedance at high frequencies were estimated.

## Data availability

Experimented data of this study are available in figshare [https://figshare.com/s/5c86ac9f0164f90f12a7], and all data generated simulation data during this study are included in the published article and Supplementary Information is available from the corresponding authors upon request.

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

## Author contributions

M.I. and K.I. conceived, designed, performed the experiments and computational studies, and organized the manuscript. T.U. and S.K. performed pouch cells fabrication and the destructive examinations. H.K. performed electrochemical analysis, and T.S. supervised the work. All authors discussed the results and commented on the manuscript.

## Competing interests

The authors declare no competing interests.
