## [Peer Review File · Nature Communications]

Operando Li metal plating diagnostics via MHz band electromagneticsREVIEWER COMMENTS

Reviewer #1 (Remarks to the Author):

This paper describes a new method for detection of lithium plating in Li-ion batteries using the high frequency electron response in the metallic phase. The technique has potential for use as a diagnostic tool in both battery research and commercial applications. The work is novel, appears robust (subject to a caveat around repeatability) and should be suitable for publication in Nature Communications following revision as outlined below.

While the concept of the work is neat, my main concern is around the lack of information provided by the authors regarding the repeatability of their measurements. No mention is made of the number of repeat tests, relatively few error bars are evident in the plots and no statements of measurement uncertainty are provided. This must be addressed before publication can be considered.

In Figure 3c there is considerable scatter in the measurement of $\text{Re}[\Delta Z_{>1\text{MHz}}]$ for the commercial LFP 18650 cell in the absence of lithium plating (i.e. under normal cycling conditions). What is the practical resolution of the technique given this relatively fluctuating baseline? Again, analysis of measurement uncertainty would be beneficial here.

The paper nicely combines two different fields – electrochemistry and electromagnetics. Unfortunately, one of the downsides of this is that relatively few readers will have the scientific background required to understand in sufficient detail all of the concepts presented. In my opinion, the authors could do a better job of explaining the mechanism of the high frequency response to the general reader. For example, the use of clearer schematic diagrams rather than loose descriptive cartoons could be considered. The manuscript would also benefit from thorough revision by a native English speaker to assist with clarity and flow of the text.

Reviewer #2 (Remarks to the Author):

Authors report on 'holy grail' of battery sensor characterization, an operando and non-destructive technique with the ability to detect lithium plating and dendrite growth in full commercial lithium-ion battery cells degraded under rapid charging. The paper is hard to read, non-scientific and seems incomplete, as it often lacks references to statements such as 'When operating at frequencies way above the conventional EIS frequency band, the battery behaves as an electric double-layer capacitor'. Which frequencies are these and under which circumstances? In Figure 1, the authors discuss a high-frequency impedance model (EIS models are typically not frequency dependent but are showing different features depending on the measured frequencies) without explanation of its origin and the rationale for its build-up. On the other hand, the reported electromagnetic model in Figure 2 heavily

relies on 'skin effect' which seems complex and is also not explained in detail, and computational analysis, the origin and limits of which are unclear. From Figure 2, it seems that the authors also suggest a change in battery design related to the placement of terminals, necessary for the suggested technique. Terminology such as 'diagonal terminal model' has been used without its definition or introduction. A number of sentences are highly confusing such as 'response of high-frequency electrons, which are minority carriers in batteries' – mixed ionic/electronic conductivity is necessary in any battery electrode, electrons and ions move at any frequency, only their movement is related to different time/lengthscales (e.g. bulk or interfaces).

Reviewer #3 (Remarks to the Author):

The manuscript reports on a diagnostic operando method for the lithium plating effects, which appears quite feasible.

1. The effects of the lithium plating appear similar to the high frequency behavior often observed below 0.1 MHz, which is modeled by LR "parallel" circuit, e.g. in 10.1016/j.jpowsour.2022.231814. The origin of this behavior has not been clearly explained. Can the authors explain this effect? The effect would increase with frequencies and overwhelm the lithium plating effects.

The authors wrote that the response cannot be modeled as ECM and named it "diffuse" impedance. Do they mean distributed elements? Distributed elements can also be modeled as transmission lines.

Would or should the technique be applied to the individual cells in the module or pack or to the modules and packs? Electromagnetic effects should arise from the electrical connections and possibly from the aluminum pouches.

COMSOL simulation file provided does not contain the impedance responses, which is essential.

Overall the language and scientific writing should be improved. Some examples are as follows.

44: categlize : typo

45-47: Monitoring internal growth of Li-metal deposition is critical for evaluating its state of safety (SOS) because it has been identified as a factor that significantly reduces(?) the starting temperature of thermal runaway: need rewriting. What is reducing?

48-50: Nonetheless(?), the battery safety is a concern not only for secondary(?) users but also for ancillary industries like transportation and storage for battery reusing ... : Please check the expression, which is not clear.

49-50: who cannot confirm internal Li-metal deposition by destructive processes...: Who can?

51-52: within common understand of the battery safety. : Need improvement in writing.

53-54 that do not rely on physical examination or historical narratives(?)

55: ..., which have been little attention in electrochemical devices.

... Overall substantial rewriting is needed!

"floated electrochemically(?) from the graphite" : This expression is not clear in meaning.

87 $Z_{sep} = C_{hf} R_{hf}$: The authors may mean RC parallel circuit. RC product is not the impedance , however.

88: that serves as the source of the displacement current, with electron(?) as the carrier(?). : C_{hf} as parallel to the ionic path. Displacement current does not need to specify carriers. Capacitance magnitudes matter.

96: ..in the high-frequency is appeared

100: ... skin-depth is depend on ...

192: ... carbon newtrality

533 with t the

Fig. 3c nominal current ratio -> nominal C-rate. Box the legend to be distinguished from the data points.

Supp. Table 1: Check the standard charge current for NCM.

COMSOL simulation does not have impedance calculation.

What is the "diffuse impedance"

282: ...the conventional ECM cannot be used because fitting must be conducted across an acceptable frequency range: The meaning is not clear and appears non-scientific.

Dear Editors and Reviewers,

We greatly appreciate your effort for reviewing the manuscript titled " Operando safety diagnosis for Li-ion battery via MHz band electromagnetics"

Thank you for your valuable feedback. Based on the constructive comments received from the reviewers during the first reviewing process, we have thoroughly revised and improved the manuscript. The overview of the revision is addressed as follows:

- Clear explanation of electromagnetic principles and behaviour:
We have reworked the explanations of high-frequency electromagnetic behaviour in a battery to ensure they are accessible and comprehensible to experts in electrochemical technologies.
- Comprehensive explanation of mechanisms, including high-frequency equivalent circuit model:
The mechanisms, including the equivalent circuit model, have been explained in a more detailed and coherent manner to enhance the reader's understanding. References 43-46 regarding the ECM are added for this revision.
- Quantification of measurement accuracy:
We have provided a more detailed quantification of the measurement accuracy to in the experimental result.
- English language proofreading:
The manuscript has revised with English language proofreading by Nature Service to ensure accuracy and clarity of the manuscript.

The explanation of the principles has been significantly revised, resulting in almost the entire text being modified. The comments and responses are summarized in this document. Please refer the comment and reference information for your review. By addressing these areas, we believe the revised manuscript aligns more effectively with the expectations of both the readership and the reviewers. We appreciate your time and effort in reviewing our revised manuscript and hope that the revised version meets the desired standards for publication.

Sincerely,
Masanori Ishigaki, Toyota Central R&D Labs., IINC.

Continue to next page

Answer for comments from Reviewer #1

#1-1 / The technique has potential for use as a diagnostic tool in both battery research and commercial applications. The work is novel, appears robust (subject to a caveat around repeatability) and should be suitable for publication in Nature Communications following revision as outlined below.

Answer : Thank you for your positive feedback. Since the beginning of our research, we have been engaged in the pursuit of principles that are beneficial to society, so this comment on the results is a great honor for the research teams. Based on valuable feedback from the reviewer, we have revised manuscript for contributing to the battery community. We greatly appreciate your review for improving paper quality.

Continue to next page

#1-2 / While the concept of the work is neat, my main concern is around the lack of information provided by the authors regarding the repeatability of their measurements. No mention is made of the number of repeat tests, relatively few error bars are evident in the plots and no statements of measurement uncertainty are provided. This must be addressed before publication can be considered.

Answer : Thank you for your comment. We updated experimental results shown in Fig.3 and Fig.4 by referring accuracy and repeatability test that summarized in Extended Data Fig.6. In addition, the condition of accuracy verification is noted in Method. The result of standard deviation is 1.61 m Ω , which is 0.5% of the 18650-battery impedance at 1MHz. This result is updated in Fig.3 as error bar, which is ± 2.3 m Ω of $\text{Re}[\Delta Z]$.

Please refer to the following information for your review. Thank you.

Page 9, line 165-168 : “In order to ensure the accuracy of $\text{Re}[\Delta Z]$, the variability of contact resistance and the stray loss occurred in surrounding metal need to be carefully minimized. Extended data Fig. 6a summaries practical accuracy of the measurement method as standard deviation.”

Page 13, line 220-222 : “The high-frequency impedance is measured using a two-port network analyser (Keysight E5061B, $\pm 7\text{V}$ input, 100kHz to 2GHz) and the shunt-through measurement technique that is most accurate method for low resistance measurement (averaged eight times)”.

Please review the revised figure 3 in page 25 and 26, with extended data fig.6 in page 33.

Continue to next page

Figure.3

“b,c Measured battery impedance change $Re[\Delta Z]$ at 20 °C and SOC = 60 %. The standard deviation at 1 MHz is $\pm (1.61^2 + 1.61^2)^{0.5} = \pm 2.27 \text{ m}\Omega$, based on the extended data in Fig. 6.”

Extended data Fig.6

“a,b Characteristics of the network analyser using a shunt-through measurement method. The standard deviation and coefficient of variation are measured by repeatedly removing and reinserting the battery socket five times.”

Continue to next page

#1-3 / In Figure 3c there is considerable scatter in the measurement of $\text{Re}[\Delta Z_{1\text{MHz}}]$ for the commercial LFP 18650 cell in the absence of lithium plating (i.e. under normal cycling conditions). What is the practical resolution of the technique given this relatively fluctuating baseline? Again, analysis of measurement uncertainty would be beneficial here.

Answer : Thank you for your comment. Similar to the response to the comment #3-1, we have added error bars reflecting the results from Extended Data Fig. 6 to Fig. 3c. As a result, it is understood that the fluctuations near zero in the results of the normal cycling conditions are mostly due to measurement instrument-dependent variations. These errors minorly include thermal fluctuations of the measuring instrument, but are mostly attributed to the influence of contact resistance. In the conclusion of this paper, there is a measurement error (standard deviation) is around $\pm 2.3\text{m}\Omega$. However, it is becoming possible to improve this measurement accuracy more than decades by developing the jig through S-parameter measurements described in the Methods section. We will discuss and publish this new technique with measurement instrument manufacturers, after this manuscript publication.

For the most relevant revised text, please refer to the following information for your review. Thank you.

Page 9, line 165-168 : “In order to ensure the accuracy of $\text{Re}[\Delta Z]$, the variability of contact resistance and the stray loss occurred in surrounding metal need to be carefully minimized. Extended data Fig. 6a summaries practical accuracy of the measurement method as standard deviation.”

Page 13, line 220-222 : “The high-frequency impedance is measured using a two-port network analyser (Keysight E5061B, $\pm 9\text{V}$ input, 100kHz to 2GHz) and the shunt-through measurement technique that is most accurate method for low resistance measurement (averaged eight times).”

Page 33, line 639-641 : “Characteristics of the network analyser using a shunt-through measurement method. The standard deviation and coefficient of variation are measured by repeatedly removing and reinserting the battery socket five times.”

Continue to next page

#1-4 / The paper nicely combines two different fields – electrochemistry and electromagnetics. Unfortunately, one of the downsides of this is that relatively few readers will have the scientific background required to understand in sufficient detail all of the concepts presented. In my opinion, the authors could do a better job of explaining the mechanism of the high frequency response to the general reader. For example, the use of clearer schematic diagrams rather than loose descriptive cartoons could be considered.

Answer : We greatly appreciate your perspective for this manuscript. We've received similar feedback from other reviewers as well, and as a result, the section titled "High-frequency electromagnetic behaviour in Li-metal deposited LiB." is significantly revised. As you've pointed out, the revised manuscript now includes explicit changes that illustrate electromagnetic behaviour in the battery. In addressing the two electromagnetic behaviour that crucial for comprehension in this manuscript – namely, the skin effect and the proximity effect – we've added two new figures. Figure 1c outlines how the skin effect occurs within the multi-layered copper, graphite, and electrolyte, using a simplified simulation model to illustrate the generation of surface current flowing at the graphite interface. We've incorporated an equivalent circuit understanding corresponding to this skin effect into Figure 1b, by introducing it into the equivalent circuit model (ECM) that traditionally used in electrochemistry at DC to low frequencies. Additionally, Figure 2a demonstrates the proximity effect that occurs in the battery structure, including the terminals, through variations in current density. Conversely, for those in the field of electrical engineering who might find difficulty to grasp the Li-metal deposition, we've retained the cartoons depicting the internal battery structure that follow the description in Figure 1b.

For the most relevant revised text, please refer to the following information.

Page 6, line 90-107 : “In contrast, at frequencies above 100 kHz, the electromagnetic effects based on Maxwell’s equations become significant ... As a result, the i_{xy} flow is concentrated on the thin metal surface by the growth of Li-metal plating, which has a larger conductivity than graphite.”

Page 7, line 110-116 : The high-frequency magnetic field H_z induces another current concentration against the i_{xy} that diffuses on the graphite surface. ... those of Li metal (Fig. 2b and Extended Data Fig. 2a-c).

Continue to next page

Figure. 1c

“c Current concentration by the skin effect in multilayered, different conductive materials in electromagnetic FEM analysis. The counter of the current density in the xy surface of the edge of the cylinder model visualises the current distribution in the z and xy directions by electromagnetic FEM analysis. The CR-Model B including R_{Li-M} in (b) would appear as a low-resistance channel against the surface current in the xy direction.”

Figure.2a

“a Current concentration by the proximity effect. The copper model imitates the laminate cell shown in (b) and the extended data in Fig. 3. The vector line shows the current distribution in the xy plane.”

Figure. 1a,d

“a Graphical model of Li-metal plating with a low-frequency electrochemical reaction. The intercalation reaction is interfered with by Li-metal plating on the graphite surface.”

“d Graphical model of Li-metal plating with a MHz electromagnetic reaction. The electron current distributes to the xy direction on the graphite surface where the Li-metal plate grows. This surface current can pass through Li-metal via a polarisation layer (C_{SEI}) as a displacement current.”

#1-5 / The manuscript would also benefit from thorough revision by a native English speaker to assist with clarity and flow of the text.

Answer: Thank you very much for your advice. We used Nature Author Service (English proofread) to make our manuscript better and clear.

Continue to next page

Answer for comments from Reviewer #2

#2-1 / The paper is hard to read, non-scientific and seems incomplete, as it often lacks references to statements such as ‘When operating at frequencies way above the conventional EIS frequency band, the battery behaves as an electric double-layer capacitor’. Which frequencies are these and under which circumstances?

Answer : We truly appreciate your feedback. By this comment, we've been able to identify areas where the first manuscript was lacking and have subsequently incorporated quantitative and technical descriptions to provide a more comprehensive understanding of the technology. In the revised manuscript, we've carefully categorized the frequency ranges handled by this manuscript. The conventional EIS frequency range is referred to as the kHz range, while the MHz range, which highlights distinct features in this measurement, is now designated as the high-frequency range. Although the section "High-frequency electromagnetic behaviour in Li-metal deposited LiB" has significant revisions to explain the principles by other feedbacks, the specific term of this comment is removed from the revised manuscript. However, the responses for this comment are now described in the revised Figure 1b. We kindly ask you to review the following description for your confirmation. Thank you very much.

Page 23, line 485-490 : “b Equivalent circuit model (ECM) of the anode electrode with frequency-dependent behaviours. R_m and L_m represent terminal impedance, and R_{cu} and R_{gr} represent resistance of ... the CR-Model B including R_{Li-M} is located parallel to $R_{gr(xy)}$, which describes graphite resistance in the xy direction.”

Please review the revised figure 1b in page 23.

Continue to next page

Figure. 1b

“b Equivalent circuit model (ECM) of the anode electrode with frequency-dependent behaviours. R_{tm} and L_{tm} represent terminal impedance, and R_{cu} and R_{gr} represent resistance of the anode collector and the active material. CR-Model A consisting of C_{dl} , R_{ct} and Z_w describes the charge transfer impedance, and CR-Model B consisting of C_{SEI} and R_{SEI} describes the impedance of the SEI layer. The resistance of Li metal R_{Li-M} appears as a series resistance to CR-Model B. CR-Model B, including R_{Li-M} , is located on the surface side of the CR ladder network in the kHz band. In MHz, the CR-Model B including R_{Li-M} is located parallel to $R_{gr(xy)}$, which describes graphite resistance in the xy direction.”

Continue to next page

#2-2 / In Figure 1, the authors discuss a high-frequency impedance model (EIS models are typically not frequency dependent but are showing different features depending on the measured frequencies) without explanation of its origin and the rationale for its build-up.

Answer : Thank you very much for your comment. We understand that the ECM (Equivalent Circuit Model) of a battery is a model that uses the characteristics of circuit networks (typically CR parallel) to simulate the frequency response of a battery. In the initial version of the manuscript, we tried to simply explain that the primary responses are different in each frequency domains, but the explanation was inadequate. In the revised manuscript, we modified the ECM in Figure 1b that can demonstrate behavior at high frequencies more clearly. As you know, the ECM has been reported to represent depth-directional responses occurring at around 1 kHz by transforming the 1D Randles model into a Distributed Model. In this paper, the electromagnetic effect revolves around new paths of surface current at high frequencies in the MHz range. Figure 1c explains the mechanism of surface current generation, while illustrating this response with an ECM in the MHz range in Figure 1b. This model represents the diffusing current on the graphite surface due to the skin effect, allowing for describing of changes in the current path at the active material interface, which causes a conductivity transformation (graphite to Li-Metal). These modifications have led to an entire revision of the section "High-frequency electromagnetic behaviour in Li-metal deposited LiB."

We kindly ask you to review the following description for your confirmation.

Page 5, line 70-107 : "Here, the detection mechanism is attributed to the extended equivalent circuit model (ECM) shown in Fig. 1 ... is concentrated on the thin metal surface by the growth of Li-metal plating, which has a larger conductivity than graphite."

Please review the revised figure 1b in page 23.

Continue to next page

Figure. 1b

“b Equivalent circuit model (ECM) of the anode electrode with frequency-dependent behaviours. R_{tm} and L_{tm} represent terminal impedance, and R_{cu} and R_{gr} represent resistance of the anode collector and the active material. CR-Model A consisting of C_{dl} , R_{ct} and Z_w describes the charge transfer impedance, and CR-Model B consisting of C_{SEI} and R_{SEI} describes the impedance of the SEI layer. The resistance of Li metal R_{Li-M} appears as a series resistance to CR-Model B. CR-Model B, including R_{Li-M} , is located on the surface side of the CR ladder network in the kHz band. In MHz, the CR-Model B including R_{Li-M} is located parallel to $R_{gr(xy)}$, which describes graphite resistance in the xy direction.”

Continue to next page

#2-3 / On the other hand, the reported electromagnetic model in Figure 2 heavily relies on 'skin effect' which seems complex and is also not explained in detail, and computational analysis, the origin and limits of which are unclear.

Answer : We greatly appreciate your comment capturing the essence of the paper. Your feedback and similar questions raised by other reviewers (#1-4) led us to clear direction to revise and provide a detailed explanation of how the skin effect and proximity effect contributes this metal detection mechanism within the battery structure. In the revised paper, we've employed a simplified multilayer cylindrical model in Figure 1c to clearly explain how current concentration occurs at material boundaries and electrode edges with the material conductivity information. Furthermore, within the ECM in Figure 1b, we've introduced additional modeling considering the changes in high-frequency current paths due to the Li-metal plating. These changes entirely improved the "High-frequency electromagnetic behaviour in Li-metal deposited LiB" section, as aligning with the essence of the manuscript and enhancing the comprehension of the presented principles.

Please review the following description for your confirmation (same as #2-2). Thank you very much.

Page 5, line 70-107 : "Here, the detection mechanism is attributed to the extended equivalent circuit model (ECM) shown in Fig. 1 ... is concentrated on the thin metal surface by the growth of Li-metal plating, which has a larger conductivity than graphite."

Please review the revised figure 1c in page 23.

Continue to next page

Figure. 1c

“c Current concentration by the skin effect in multilayered, different conductive materials in electromagnetic FEM analysis. The counter of the current density in the xy surface of the edge of the cylinder model visualises the current distribution in the z and xy directions by electromagnetic FEM analysis. The CR-Model B including R_{Li-M} in (b) would appear as a low-resistance channel against the surface current in the xy direction.”

Continue to next page

#2-4 / From Figure 2, it seems that the authors also suggest a change in battery design related to the placement of terminals, necessary for the suggested technique. Terminology such as ‘diagonal terminal model’ has been used without its definition or introduction.

Answer : Thank you for your feedback. The previous explanation was information-rich and may have caused confusion. I truly appreciate your comment. The purpose of showing multiple tab configurations was to practically understand the relation between the sensitivity and the battery structures. In the revised manuscript, we've made modifications that can capture the essence of structural feature in a simple and clear manner.

The factors influencing sensitivity variations due to tab positions are attributed to electromagnetic behaviour, the proximity effect. To address this, we've introduced the impact of the proximity effect in the battery structure through current concentration contour map in Figure 2a. This provides an easy visualization of how the placement of tabs can alter the edges where current concentrates. As an example, Figure 2e illustrates that different tab placements result in varying edge lengths affected by the proximity effect, and longer edges results higher sensitivity, as indicated by the simulation results. Furthermore, we've added measurement results for $\text{Re}[Z_{1\text{MHz}}]$ in three different types of 18650 batteries to Extended Data Fig. 5d. This additional data demonstrates that batteries with longer edge lengths indeed exhibit larger absolute values of high-frequency resistance and greater changes in high-frequency resistance due to Li-metal plating.

For the most relevant revised text, please refer to the following information.

Page 6, line 108-138 : " The magnetic field H_z caused by the i_{xy} represents a source of Li-metal detection. Figure. 2 demonstrates the response of the real part of the impedance ... The intensity of the negative correlation increases when the terminals are located on the far side since it drives H_z stronger by a longer edge for the i_x ."

Please review the revised figure 2a in page 24.

Figure.2a

“a Current concentration by the proximity effect. The copper model imitates the laminate cell shown in (b) and the extended data in Fig. 3. The vector line shows the current distribution in the xy plane.”

Continue to next page

#2-5 / A number of sentences are highly confusing such as ‘response of high-frequency electrons, which are minority carriers in batteries’ – mixed ionic/electronic conductivity is necessary in any battery electrode, electrons and ions move at any frequency, only their movement is related to different time/lengthscales (e.g. bulk or interfaces).

Answer : We appreciate your feedback. As you mentioned, ions and electrons exhibit frequency-dependent responses as they flow through the battery. we acknowledge that the term "minority" can lead to confusion due to poor word management. Thank you for your pointing out. In the revised manuscript, we've replaced the corresponding expression with "minimal attention in previous demonstrations" to clarify the concept. Additionally, to enhance the supplementary understanding of battery responses in the MHz range, we've added experimental results to Extended Data Fig. 1. Regarding the clarification for the added experimental results, we kindly request to refer to the comment #3-1.

Page 2, line 13-15 : “Here, we show a direct Li-metal detection technology focusing on electromagnetic behaviour, which has received minimal attention in previous demonstrations of electrochemical devices.”

Additional extended data Fig. 1 verified the impedance response of the electrolyte layer including the separator ($\text{Re}[Z_{elec}]$) transit from a specific value to zero by frequency increase. This indicates the behavior of a CR model of electrolyte shifts from a resistance-dominant to a capacitance-dominant behavior in high-frequency. In order to describe this response in the electromagnetic FEM analysis presented in Figure 2, we've treated the electrolyte layer as a polarizable response (dielectric material with low conductivity) for modeling convenience as shown in extended data Fig.2.

Please review the revised extended data Fig. 1 in page 28, and extended data Fig.2.

Continue to next page

Extended data Fig.1a-c

“**a** A cell configuration for verifying the high-frequency response of the Z_{elec} . The cell is designed as an electric double-layer capacitor that uses the same electrolyte and separator as the laminate cell shown in the extended data Fig. 3. The thickness of the electrolyte is changed by stacking the separator. **b,c** Frequency characteristics of the measured real part of the impedance $Re[Z]$ and estimated equivalent circuit of Z_{elec} . R_{elec} represents the electrolyte resistance, and C_{elec} represents the geometrical stray capacitance between the collector plates. The R_{elec} is proportional to the thickness of the electrolyte up to 4 MHz. However, $Re[Z]$ converges to the same value from 70 MHz with increasing value by the skin effect and the proximity effect. This behaviour can be modelled by the CR model shown in (c), which uses the conductivity and relative dielectric constant as $\sigma_{elec} = 0.01$ [S/m] and $\epsilon_{elec} = 81$, respectively, which are used in other simulations, such as the extended data in Fig. 2. The estimated CR model has a cut-off of approximately 10 MHz, and $Re[Z_{elec}]$ can converge to zero by increasing the frequency. In addition, R_{elec} can be sufficiently small by increasing the cross-sectional area of S_{elec} in a large-capacity battery. Therefore, the high-frequency measurement can deal with ionic degradation as negligible, and even Z_{elec} might change its resistance by degradation.”

Continue to next page

“c Design of the lamination slices and parameters. The graphite layer was sliced into 10 layers, and special conductivity can be applied to the top two graphite layers to emulate Li-metal plating.”

Extended data Fig.2c

Continue to next page

Answer for comments from Reviewer #3

#3-1 / The manuscript reports on a diagnostic operando method for the lithium plating effects, which appears quite feasible.

Answer : We deeply appreciate your valuable comments, and we are grateful for your input. Visualizing battery safety is a crucial technology for both manufacturers and users. Through this revisions, we aim to make the technical principles clearer for contributing to the battery community. We sincerely appreciate your review for improving paper quality.

Continue to next page

#3-2 / The effects of the lithium plating appear similar to the high frequency behavior often observed below 0.1 MHz, which is modeled by LR "parallel" circuit, e.g. in 10.1016/j.jpowsour.2022.231814. The origin of this behavior has not been clearly explained. Can the authors explain this effect? The effect would increase with frequencies and overwhelm the lithium plating effects.

Answer : We sincerely appreciate your detailed feedback, including the reference paper. Inductance is a circuit component that simulates magnetic behavior, but unfortunately, our mechanism cannot be adequately explained using parallel inductance. The parallel inductance you referred may represent the increase in real-part impedance due to the skin effect in a way of simple fitting. However, it lacks electromagnetic fundamentals and isn't commonly used even in regular motor or magnetic device models that have to deal the skin effect for its design. We understand a distributed constant circuit is more appropriate to treat the frequency-dependent behaviour. Therefore, the revised manuscript modified these EMC by using multiple resistance and capacitor networks as shown in Figure 1b. We kindly ask you to review the revised ECM in Figure. 1b with following description.

Page 23, line 482-496 : "Figure 1 | Frequency response of the Li-ion battery including electromagnetic response in the MHz... layer (C_{SEI}) as a displacement current."

Page, 28, line 565-577 : " Extended Data Fig. 1 | Verification of the ECM for EIS. a-c The ECM of electrolyte Zelec at high frequency. ... degradation as negligible, and even Zelec might change its resistance by degradation."

Continue to next page

Figure. 1b

“b Equivalent circuit model (ECM) of the anode electrode with frequency-dependent behaviours. R_{tm} and L_{tm} represent terminal impedance, and R_{cu} and R_{gr} represent resistance of the anode collector and the active material. CR-Model A consisting of C_{dl} , R_{ct} and Z_w describes the charge transfer impedance, and CR-Model B consisting of C_{SEI} and R_{SEI} describes the impedance of the SEI layer. The resistance of Li metal R_{Li-M} appears as a series resistance to CR-Model B. CR-Model B, including R_{Li-M} , is located on the surface side of the CR ladder network in the kHz band. In MHz, the CR-Model B including R_{Li-M} is located parallel to $R_{gr(xy)}$, which describes graphite resistance in the xy direction.”

Continue to next page

Extended data Fig. 1a-c

“a A cell configuration for verifying the high-frequency response of the Z_{elec} . The cell is designed as an electric double-layer capacitor that uses the same electrolyte and separator as the laminate cell shown in the extended data Fig. 3. The thickness of the electrolyte is changed by stacking the separator. b,c Frequency characteristics of the measured real part of the impedance $Re[Z]$ and estimated

equivalent circuit of Z_{elec} . R_{elec} represents the electrolyte resistance, and C_{elec} represents the geometrical stray capacitance between the collector plates. The R_{elec} is proportional to the thickness of the electrolyte up to 4 MHz. However, $Re[Z]$ converges to the same value from 70 MHz with increasing value by the skin effect and the proximity effect. This behaviour can be modelled by the CR model shown in (c), which uses the conductivity and relative dielectric constant as $\sigma_{elec} = 0.01$ [S/m] and $\epsilon_{elec} = 81$, respectively, which are used in other simulations, such as the extended data in Fig. 2. The estimated CR model has a cut-off of approximately 10 MHz, and $Re[Z_{elec}]$ can converge to zero by increasing the frequency. In addition, R_{elec} can be sufficiently small by increasing the cross-sectional area of S_{elec} in a large-capacity battery. Therefore, the high-frequency measurement can deal with ionic degradation as negligible, and even Z_{elec} might change its resistance by degradation.”

Continue to next page

#3-3 / The authors wrote that the response cannot be modeled as ECM and named it "diffuse" impedance. Do they mean distributed elements? Distributed elements can also be modeled as transmission lines.

#3-11 / What is the "diffuse impedance"

Answer : Thank you for your clarification. "diffuse impedance" in the initial version should be replaced to "diffusion impedance." In the revised manuscript, taking into consideration feedback from other reviewers, the Equivalent Circuit Model (ECM) in Figure 1b was carefully revised to illustrate the detection principle. The ECM now starts with the simplest Randles circuit model and gradually transitions to a transmission line-type model that can represent the properties in the stacking direction from intermediate stages around kHz. Moreover, for the MHz frequency range, we've included high-frequency ECM that reasonably demonstrates how the surface current at the anode interface responds to changes in material conductivity due to Li-metal plating. Additionally, additional experimental result in extended data Fig. 1 represent the dominant components of impedance in the MHz range. This additional verification was conducted using an electrical double-layer cell with the same electrolyte and separator material. The experiment changed thickness of the separator to observe electrolyte behaviour in MHz band. When observing the results of $\text{Re}[Z]$, the resistance increases proportionally with the total number of separators in range of 100 kHz. At frequencies exceeding 10 MHz, a significant resistance increase due to the skin effect is observed. Notably, the value of $\text{Re}[Z]$ converges into one line in 50-100MHz, regardless of separator thickness. This response can be explained by the real-part impedance of the CR parallel circuit, which theoretically estimated as $\text{Re}[Z_{elec}]$, where the C is approximated as the impedance of a simplified parallel plate capacitor with the dielectric constant between collector distances and material properties. We kindly ask you to review the revised ECM in Figure. 1b with following description (same as #3-2).

Page 23, line 481-495 : "Figure 1 | Frequency response of the Li-ion battery including electromagnetic response in the MHz band. a Graphical model of Li-metal plating with a low-frequency ... where the Li-metal plate grows. This surface current can pass through Li metal via a polarisation layer (C_{SEI}) as a displacement current."

Continue to next page

#3-4 / Would or should the technique be applied to the individual cells in the module or pack or to the modules and packs? Electromagnetic effects should arise from the electrical connections and possibly from the aluminum pouches.

Answer : Thank you for your comment. As you pointed out, we carefully deal the influence of external metals in overall experiment. Although we added information regarding to the effect of external metals, the impact of external metals is effectively removed in this verification while the measured impedance is calculated as $\text{Re}[\Delta Z]$ for the detection. In term of extension to module or pack measurement, it is theoretically possible to apply the principle for multiple cells together. However, within the measurement environment (Keysight E5061B), the voltage range of the network analyzer is limited to 9V. Therefore, in practical terms, two cells are the maximum that can be accommodated. An additional DC cut capacitor may solve this constraint for multiple cell measurement. Furthermore, for impedance measurements using the shunt-through method, the recommended range for absolute impedance value is limited to 50Ω. Therefore, accuracy concern arises about when a large number of cells are connected in series.

To communicate these considerations, we've made the following addition comment. Thank you very much.

Page 8, line 146 : “However, its results may be affected by surrounding metal changes, e.g., deformation of metal enclosures.”

Page 9, line 165-169 : “In order to ensure the accuracy of $\text{Re}[\Delta Z]$, the variability of contact resistance and the stray loss occurred in surrounding metal need to be carefully minimized. Extended data Fig. 6a summaries practical accuracy of the measurement method as standard deviation.”

Page 13, line 220-222 : The high-frequency impedance is measured using a two-port network analyser (Keysight E5061B, ±9V input, 100kHz to 1.5GHz) and the shunt-through measurement technique that is most accurate method for low resistance measurement (averaged eight times).

Continue to next page

#3-5 / COMSOL simulation file provided does not contain the impedance responses, which is essential.

#3-10 / COMSOL simulation does not have impedance calculation.

Answer : Thank you for your comment. We would like to answer two comment regarding to the COMSOL simulation. We understand that the first comment (#3-5) pertains to the lack of absolute impedance values, and the second comment (#3-10) addresses the lack of information about impedance calculation methods in COMSOL analysis.

Regarding #3-5, we have addressed this comment by adding the output results of impedance from COMSOL to extended data Fig. 2f, g, and h. From this result, phase or imaginary impedance may include some response against the metal plating. However, we focused on the change of $\text{Re}[Z]$, which is potentially easy to measure a simplified operand sensor for engineering perspective.

As for the calculation method mentioned in #3-10, we used RF module provided by COMSOL for impedance calculation, which allows for calculation of both the absolute value and phase of impedance between input and output ports.

To address this comment, we have added the following information to the relevant section as you've suggested.

(for #3-5) Page 29, line 591-592 : “f,g,h Overall simulation results. ΔZ is calculated by the difference between (d) and (e).”

(for #3-10) Page 14, line 246-249 : “In this analysis, the COMSOL model utilized RF module which can calculate impedance of the model. In this model, lumped ports were installed in predetermined locations, and loss and the real part of impedance were determined by inducing a 1 A constant current.”

Continue to next page

Extended data Fig.2 f-h

“f,g,h Overall simulation results. ΔZ is calculated by the difference between (d) and (e).”

Continue to next page

#3-6 / Overall the language and scientific writing should be improved. Some examples are as follows.

Answer : We sincerely apologize for any inconvenience by our any inadvertent errors. In the revised manuscript, we have taken strong measures to ensure grammar accuracy. Especially regarding the list below, we have made the following corrections for your reference.

-44: categlize : typo → deleted.

-45-47: Monitoring internal growth of Li-metal deposition is critical for evaluating its state of safety (SOS) because it has been identified as a factor that significantly reduces (?) the starting temperature of thermal runaway: need rewriting. What is reducing?

Page 3, line 46 : Monitoring the internal growth of Li-metal deposition is critical for evaluating state of safety (SOS) of the LiBs, because such growth has been identified as a factor of reducing thermal runaway temperature of the LiBs

-48-50: Nonetheless(?), the battery safety is a concern not only for secondary(?) users but also for ancillary industries like transportation and storage for battery reusing ... : Please check the expression, which is not clear.

Page 3, line 49 : The battery safety is a concern not only for battery application users but also for ancillary industries such as transportation and storage for battery reusing and recycling.

-49-50: who cannot confirm internal Li-metal deposition by destructive processes...: Who can?

Page 3, line 51 : where the internal Li-metal deposition cannot be measured by destructive processes

-51-52: within common understand of the battery safety. : Need improvement in writing. → deleted.

-53-54 that do not rely on physical examination or historical narratives(?)

Page 4, line 54 : that does not rely on physical examination or usage history.

-55: ..., which have been little attention in electrochemical devices.

Page 4, line 55 : We have developed a direct metal detection mechanism by utilising a high-frequency electromagnetic response in LiB, demonstrating an approach that has received little attention in electrochemical devices.

-96: ..in the high-frequency is appeard → deleted.

-100: ... skin-depth is depend on ... → deleted.

-192: ... carbon newtrality

Page 12, line 217 :advance their carbon neutrality.

-533 with t the → deleted.

Continue to next page

#3-5 / "floated electrochemically(?) from the graphite" : This expression is not clear in meaning.

Answer : Thank you for your comment. We inadvertently simplified the description about the lithium-metal deposition that cannot return from metal to ions and is no longer usable for electrochemical reactions. As we provided a more detailed explanation of the principle, the relevant description has been removed from the revised manuscript. In Figure 1b, the lithium-metal plating is now described as a part of high-frequency current path, coated by the SEI that modeled as C_{SEI} in MHz band. For reference, we've included the relevant modification to your feedback below:

Page 6, line 103-107 : "The Li-metal plating was deposited on the way of i_{xy} . The i_{xy} can trough the Li-metal plating even though it forms the SEI, since the SEI model shown as CR-Model B could transform its reaction from resistive R_{SEI} to capacitive C_{SEI} in the MHz frequency band. As a result, the i_{xy} flow is concentrated on the thin metal surface by the growth of Li-metal plating, which has a larger conductivity than graphite."

#3-6 / 87 $Z_{sep}=C_{hf} R_{hf}$: The authors may mean RC parallel circuit. RC product is not the impedance , however.

Answer : Thank you for your comment. We carefully modified the section "High-frequency electromagnetic behaviour in Li-metal deposited LiB" to describe the behaviour of the high-frequency impedance in a battery structure. The pointed-out impedance equation has been deleted and modified accurately represented in Figure 1 and Extended Data Fig.1, which has explained in comment #3-3. We added the correct impedance equation in Extended Data Fig.1 as reference.

Extended data Fig.1b

Continue to next page

#3-7 / 88 that serves as the source of the displacement current, with electron(?) as the carrier(?). : C_{hf} as parallel to the ionic path. Displacement current does not need to specify carriers. Capacitance magnitudes matter.

Answer : Thank you for your clarification. In the process of extensively revising the section "High-frequency electromagnetic behaviour in Li-metal deposited LiB," we have refrained from using the term " C_{hf} ." As depicted in Figure 1b, the SEI impedance components present in various locations from the negative electrode interface towards the electrolyte. we understand that the SEI obtains some dielectric properties and acts as an insulator against ionic current, but allows ions to pass through. This concept is consistent with the CR model depicting the SEI layer in reference [7, 9]. In terms of impedance characteristics, the displacement current flowing through the C component gradually increases with higher frequencies compared to the ionic (conductive) current flowing through R. I acknowledge your pointed-out that ions act as carriers in the electrolyte, therefore, displacement should not limit its carriers. For the most relevant revised text, please refer to the following information (same as #3-5).

Page 6, line 103-107 : "The Li-metal plating was deposited on the way of i_{xy} . The i_{xy} can trough the Li-metal plating even though it forms the SEI, since the SEI model shown as CR-Model B could transform its reaction from resistive R_{SEI} to capacitive C_{SEI} in the MHz frequency band. As a result, the i_{xy} flow is concentrated on the thin metal surface by the growth of Li-metal plating, which has a larger conductivity than graphite."

Continue to next page

Figure. 1b

“b Equivalent circuit model (ECM) of the anode electrode with frequency-dependent behaviours. R_{tm} and L_{tm} represent terminal impedance, and R_{cu} and R_{gr} represent resistance of the anode collector and the active material. CR-Model A consisting of C_{dl} , R_{ct} and Z_w describes the charge transfer impedance, and CR-Model B consisting of C_{SEI} and R_{SEI} describes the impedance of the SEI layer. The resistance of Li metal R_{Li-M} appears as a series resistance to CR-Model B. CR-Model B, including R_{Li-M} , is located on the surface side of the CR ladder network in the kHz band. In MHz, the CR-Model B including R_{Li-M} is located parallel to $R_{gr(xy)}$, which describes graphite resistance in the xy direction.”

Continue to next page

#3-8 / Fig. 3c nominal current ratio -> nominal C-rate. Box the legend to be distinguished from the data points.

Answer : Thank you for your comment. we have included specific current values and added information about the C-rate. Additionally, we have modified the location of the legend to avoid duplication. Your assistance is greatly appreciated.

Please confirm modified Fig.3 in page25.

Figure.3c

Continue to next page

#3-9 / Supp. Table 1: Check the standard charge current for NCM.

Answer : Thank you for very much. we have reviewed the numerical values and confirmed that the mistake was due to unit inconsistency. we appreciate your assistance. we have made the correction as follows:

Supplementary data, Sup. Table 1 in page 5 : “1.250mA → 1250mA”

#3-12 / 282: ...the conventional ECM cannot be used because fitting must be conducted across an acceptable frequency range: The meaning is not clear and appears non-scientific.

Answer : Thank you for your feedback. The setting of the fitting range was indeed unclear. We have made the necessary clarification as follows:

Page 17, line 311-308 : “The starting frequency (100 mHz) is set by the measured Cole-Cole plot, where the diffusion impedance could be regarded as a negligible value. A value of 750 kHz ($-Im[Z] > 0$) was set as the end point of the fitting frequency, where the value of the $Re[Z]$ is conventionally treated as an ohmic resistance in the Cole-Cole plot.”

End of document

REVIEWERS' COMMENTS

Reviewer #2 (Remarks to the Author):

The authors significantly improved the state of the manuscript and addressed many of the referees' comments. The work remains very complex, but also highly interesting for the battery community.

Reviewer #3 (Remarks to the Author):

The manuscript has been substantially improved.

The gap or the overlapped frequency range between MHz measurements and conventional ECM, explicitly shown in Extended Fig. 1 and 3, should be clearly explained. They cannot be the same due to the instrumentation? The present analysis relies, however, on the conventional ECM results.

Is there a reason for calling the "CR" model? "RC" model, RC ladder network is more generally used.

Scientific writing can be improved overall. Some examples are:

19: when compared to conventional methods such data-driven analysis ...

51: ... where the internal Li-metal deposition cannot be measured by destructive processes: cannot or can? Are references 18,19 relevant?

66: ... on a decrease in ionic(?) performance

82 ... to describe the concentration(?) of the ionic reaction(?) at the boundary of the graphite

Extended Data Fig. 1 a. and c: al collector, R_al-> should be Al

Answer for comments from Reviewer #3

#3-13 / The gap or the overlapped frequency range between MHz measurements and conventional ECM, explicitly shown in Extended Fig. 1 and 3, should be clearly explained. They cannot be the same due to the instrumentation? The present analysis relies, however, on the conventional ECM results.

Answer : As you pointed out, understanding the transition frequency between the conventional ECM and high-frequency ECM is important for this detection technology. I greatly appreciate receiving this feedback.

Indeed, accurate impedance measurement in wide frequency band is challenging from an engineering perspective, but the answer to your comment does not rely on this viewpoint. The answer to your comment involves a fundamental challenge in understanding the electromagnetic and electrochemical aspects in the battery, which remains future research perspectives listed in the Discussion session.

We understand that the ECM is a tool used to describe electrochemical behavior in a simple manner by utilizing circuit components. Therefore, in this study, within the frequency range where the real part of the impedance clearly increases with frequency, we consider that the behavior of the battery can be adequately described using the extended high-frequency ECM for Li-metal detection. In this idea, the measurement does not be interfered by the transition between conventional and high-frequency ECM. For example, as shown in Supplementary Fig. 4, it can be confirmed that the high-frequency resistance already significantly increases with frequency from 100 kHz in 18650-type batteries. Therefore, the extended high-frequency ECM can be applied to the overall measurement results of the network analyzer (E5061B, Keysight). In the case of small cells, as shown in Supplementary Fig. 3, the relationship between resistance and Li-metal plating can be observed at the frequency where the resistance starts to increase significantly.

To quantitatively observe the frequency range of model transition, it is necessary to clarify the high-frequency characteristics of the SEI, including dielectric material properties. As mentioned in the Discussion, this research presents another challenge but is worth pursuing not only for quantifying the frequency band of the ECM transition but also for leading to new findings for future battery design.

We hope that this paper will serve as a reference for further research, anticipating progress in research related to this insight.

Continue to next page

Please find update in the manuscript referred in below. The additional comment reasonably explains the existence of model transition, and describes the practical ways of dealing it in this manuscript.

line 139-148 : “In actual batteries, the decrease in ionic resistance is dependent on the material properties of the dielectric layers, which can be described as high-frequency characteristics of the RC-model B. In addition, in such high-frequency, the increase in electronic resistance of the graphite layer depends on the electrode structure as shown in Fig.2 and Fig.3. Therefore, transition frequency from the conventional ECM to the extended high-frequency ECM can vary in each individual battery design. In this study, within the frequency range where the real part of the impedance is measured as clear increase against frequency, we consider the behavior of the battery can be enough described as the extended high-frequency ECM for Li-metal detecting. Moreover, the RC parallel circuit behaves more capacitive by increasing the surface area, then the electromagnetic behaves gets majority of the real part impedance in high-frequency. Therefore, this Li-metal detection method works better with large capacity batteries.”

#3-12 / Is there a reason for calling the "CR" model? "RC" model, RC ladder network is more generally used.

Answer : Thank you very much for your professional comment. We modified CR to RC in all sentences. The word CR was from LCR meter with the L removed, but as you mentioned, RC is common in the circuit model.

Continue to next page

#3-13 / Scientific writing can be improved overall.

Answer : Thank you very much for your professional comment. We modified scientific writing listed as below.

19: when compared to conventional methods such data-driven analysis ...

Line 17 : This finding enables simpler diagnostics when compared to data-driven analysis because we can correlate a direct response from the electronic behaviour to the metallic material property rather changes in the ionic behaviour.

51: ... where the internal Li-metal deposition cannot be measured by destructive processes: cannot or can? Are references 18,19 relevant?

Line 47 : ...because such industries cannot evaluate the internal Li-metal deposition by destructive processes.

66: ... on a decrease in ionic(?) performance

Line 64 : which diagnose LiB safety based on a degradation in ionic reaction.

82 ... to describe the concentration(?) of the ionic reaction(?) at the boundary of the graphite

Line 77 : As an extension of the 1D model, a distributed behaviour model shown in Fig. 1c along the stack direction can be used to describe the diffusion of the lithium-ion at the boundary of the graphite.

Extended Data Fig. 1 a. and c: al collector, R_{al} -> should be AI

The modified Supplementary Fig.1 uses R_{AI}

Other red lines are mainly modified for formatting including figure modification, which is summarized in the Author checklist.

Thank you very much for your kind and worth work for this manuscript.

Sincerely,
Masanori Ishigaki, Toyota Central R&D Labs., INC.